# The visual word form area (VWFA) is part of both language and attention circuitry

Lang Chen [1,2,3,8]*, Demian Wassermann [4,8], Daniel A. Abrams[1,8], John Kochalka[1], Guillermo Gallardo-Diez[5] & Vinod Menon[1,6,7]*

While predominant models of visual word form area (VWFA) function argue for its specific role in decoding written language, other accounts propose a more general role of VWFA in complex visual processing. However, a comprehensive examination of structural and functional VWFA circuits and their relationship to behavior has been missing. Here, using high-resolution multimodal imaging data from a large Human Connectome Project cohort ($N =$ 313), we demonstrate robust patterns of VWFA connectivity with both canonical language and attentional networks. Brain-behavior relationships revealed a striking pattern of double dissociation: structural connectivity of VWFA with lateral temporal language network predicted language, but not visuo-spatial attention abilities, while VWFA connectivity with dorsal fronto-parietal attention network predicted visuo-spatial attention, but not language abilities. Our findings support a multiplex model of VWFA function characterized by distinct circuits for integrating language and attention, and point to connectivity-constrained cognition as a key principle of human brain organization.

[1] Department of Psychiatry and Behavioral Sciences, Stanford University, Stanford, CA 94394, USA. [2] Department of Psychology, Santa Clara University, Santa Clara, CA 95053, USA. [3] Neuroscience Program, Santa Clara University, Santa Clara, CA 95053, USA. [4] Parietal, Inria Saclay Île-de-France, CEA, Université Paris-Sud, 1 Rue Honoré d'Estienne d'Orves, 91120 Palaiseau, France. [5] Athena Project Team, INRIA Sophia Antipolis-Méditerranée, 06902 Sophia Antipolis CEDEX, France. [6] Department of Neurology and Neurological Sciences, Stanford University, Stanford, CA 94394, USA. [7] Stanford Neuroscience Institute, Stanford University, Stanford, CA 94394, USA. [8]These authors contributed equally: Lang Chen, Demian Wassermann, Daniel A. Abrams *email: lchen32@stanford.edu; menon@stanford.edu

The visual word form area (VWFA) is among the most studied and deliberated brain regions in the cognitive neuroscience literature[1–3]. A link between left-hemisphere ventral occipito-temporal cortex (vOTC), encompassing the VWFA, and word reading was first identified over a century ago[4], and subsequently a large body of human neuroscientific research has implicated the VWFA as the brain's putative "letterbox," processing written letters and words and transmitting this information to higher-order language regions for linguistic processing[1,5–7]. Despite extensive empirical investigation, fundamental questions remain regarding the precise functional role of the VWFA and, relatedly, the brain circuitry supporting its function.

The predominant model of VWFA function, referred to here as the language model, states that the VWFA has a specific computational role in decoding written forms of words and is considered a crucial node of the brain's language network[1,2]. Consistent with the language model of VWFA function, a large body of evidence has accumulated showing regional activation for orthographic symbols in VWFA, including letters[8] and words[6,9] compared to a range of visual control stimuli. Additional support for the language node model has been provided by studies examining structural and intrinsic functional connectivity of VWFA. For example, recent studies have shown strong profiles of white-matter[10,11] and functional connectivity[12,13] between VWFA and lateral prefrontal, superior temporal, and inferior parietal regions implicated in language-related functions. These results support the language model by suggesting that the VWFA has privileged connectivity to other nodes of the distributed language network.

A second model, which we describe as the multiplex model of VWFA function, proposes that the VWFA is not specialized exclusively for written words, and is also involved in processing other categories of visual stimuli[14–17]. While this model acknowledges that VWFA may be relatively optimized for orthographic stimuli, it further assumes that several other vOTC structures are necessarily engaged by these stimuli and together these regions comprise a distributed, large-scale circuit for processing multiple categories of visual stimuli[7,17–20]. Evidence in support of the multiplex model includes results showing that the VWFA is not only activated by written words but is also consistently activated by faces[21,22] and numerical symbols[23,24]. Moreover, findings from neuropsychological research indicate that patients with left pFG lesions reveal impairments in both word reading as well as other non-orthographic visual recognition[25].

An important additional facet of the multiplex model, based on a separate line of research, is that attentional systems play a key role in VWFA function as a means of tuning and amplifying a range of visual stimuli for use and distribution to other brain systems[17,26]. Initial evidence is provided by a whole-brain investigation of VWFA intrinsic functional connectivity[27]. Results from this study revealed a striking pattern of functional connectivity between VWFA and the fronto-parietal attention network, including intraparietal sulcus (IPS), V5/middle temporal visual area (MT+), and frontal eye fields (FEF), with relatively weak coupling with language regions, including vOTC, angular gyrus (AG), supramarginal gyrus (SMG), and inferior frontal gyrus (IFG). These findings are consistent with the multiplex model by showing that VWFA is tightly coupled to attentional systems; however, links to the brain's language network were surprisingly weak. Importantly, structural connectivity studies[11,26,28] have identified a white-matter tract connecting VWFA with the IPS, providing further support for links between VWFA and the fronto-parietal attention network.

Several methodological limitations in previous studies have precluded a more comprehensive understanding of VWFA function and circuitry. First, most previous studies examining VWFA connectivity have used relatively small sample sizes ($N <$ 30), and the stability and replicability of reported results are unknown. Second, the majority of intrinsic functional connectivity studies[12,13] have restricted their analyses to examine VWFA connectivity with predefined language regions, which have precluded a comprehensive description of VWFA connectivity with brain systems beyond the canonical language system. Third, previous VWFA connectivity studies have restricted behavioral analyses to reading measures[12,13,27] and have not examined whether VWFA connectivity is associated with non-linguistic tasks associated with visual attention, which is germane to the multiplex model of VWFA function. Finally, previous studies[10,11,13,27] have not simultaneously examined whole-brain structural and intrinsic functional connectivity of VWFA in the same participants, which would allow for a thorough characterization of VWFA circuitry and its links to behavior.

The overall goal of the current study is to test competing models of VWFA function and address unanswered questions regarding the structural and functional connectivity of VWFA (for the study overview, see Fig. 1 and Methods). Here, we use a large high-resolution multimodal imaging Human Connectome Project (HCP) cohort ($N = 313$) to address two major goals. Our first goal was to examine structural and functional connectivity of VWFA in the same cohort, with a focus on connectivity patterns with lateral prefrontal, superior temporal, and inferior parietal brain systems implicated in language function on the one hand, and fronto-parietal attention systems, on the other hand. Our second goal was to investigate brain-behavior relationships between VWFA functional and structural circuits and language and attentional abilities. Crucially, the two models of VWFA function would make differential predictions regarding structural and functional connectivity and their relationship to behavior. Specifically, the language model predicts preferential structural and functional connectivity between VWFA and left-hemisphere language nodes implicated in mapping written words to phonological and lexical representations[2,12,13], and that these connectivity patterns would be related to reading skills[29,30]. Given that the multiplex model acknowledges that VWFA plays a key role in the processing of orthographic stimuli, this model would also predict heightened connectivity with nodes of the canonical language network and relationships with reading skills. However, the multiplex model also predicts strong patterns of structural and functional connectivity between VWFA and the fronto-parietal visuospatial attention network instantiated in the IPS. Moreover, since the multiplex model posits that VWFA specialization is not restricted to language-related function, an important additional prediction is that connectivity between VWFA and the IPS will be associated with visual attention skills. A third and final goal was to provide converging evidence that the VWFA has enhanced functional connectivity to language and attention networks during reading and visuospatial attention tasks.

## Results

**Whole-brain structural connectivity of VWFA**. The first goal of the analysis was to perform whole-brain structural connectivity analysis of the VWFA using probabilistic tractography (Fig. 2). This analysis revealed three major white-matter tracts associated with the VWFA, represented in colored streamlines from one subject in Fig. 2 and enlarged in Supplementary Fig. 1. The first major white-matter tract identified for the VWFA is the inferior longitudinal fasciculus (ILF; Supplementary Fig. 1 in green), which extends ventrally and laterally from the occipital pole to the anterior temporal cortex. The ILF provides critical connections between VWFA and structures of the ventral temporal cortex,

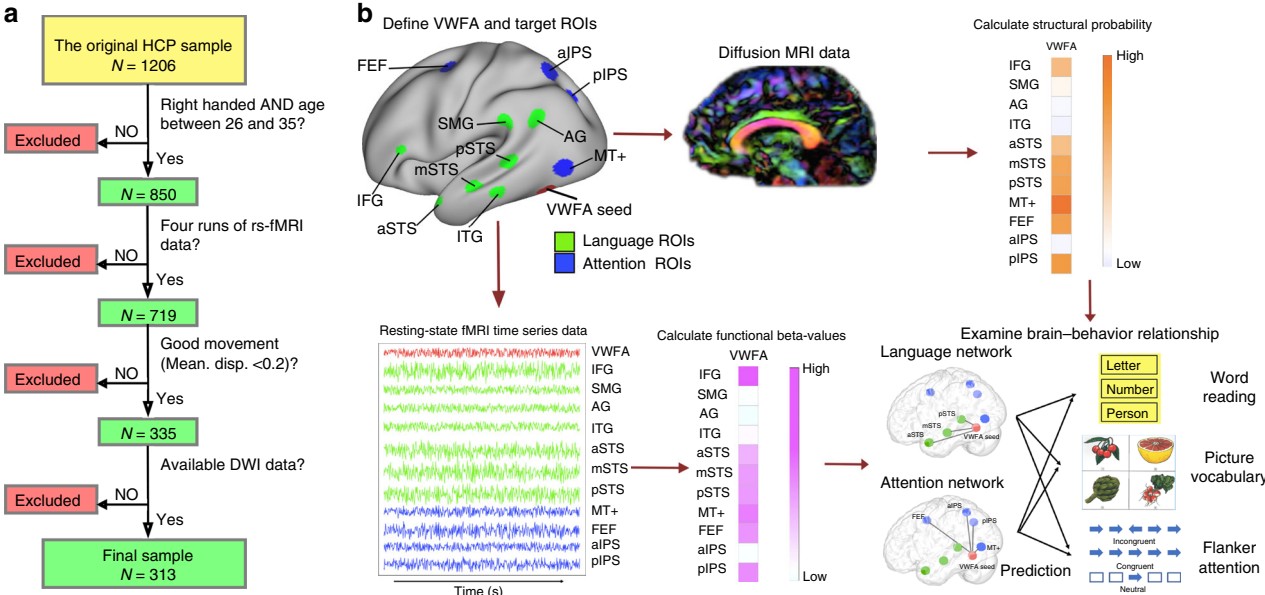

**Fig. 1 Participant selection and analysis pipelines. a** Key steps in selection of right-handed participants from the original HCP sample ($N = 1206$) to achieve a final sample size of $N = 313$; and **b** Data analysis pipeline for computing structural and functional connectivity measures and examining brain-behavior relationships (Exemplar image of picture vocabulary was adapted from Pichette et al.[30]).

including ITG and ATL. The second major white-matter tract is the superior longitudinal fasciculus (SLF; Supplementary Fig. 1 in red), which connects frontal and parietal cortices with a branch arching into the ventral surface of the occipito-temporal cortex and may provide critical structural connections between VWFA and structures of superior temporal cortex, including STS, and prefrontal cortex, including IFG and dlPFC. The third major white-matter tract identified for the VWFA is a fiber bundle that extends dorsally from the ventral surface of occipito-temporal cortex to superior and lateral parietal cortex (Supplementary Fig. 1 in blue). The orientation of this tract is consistent with findings from previous studies, which identified a white matter tract originating in the VWFA and terminating in the posterior IPS[26,28,31,32]. The structural neuroanatomy literature has produced some uncertainty regarding the nomenclature of this white matter tract, with some referring to it as ventral occipital fasciculus (vOF) based on its origins in occipital cortex, and others referring to it as temporal-parietal/superior parietal lobule (TP-SPL) tracts based on its origins in inferior temporal cortex[28]: we refer to the tracts identified in our study as the vOF/TP-SPL. To highlight the similarity between the vOF/TP-SPL identified in our structural connectivity analysis (Supplementary Fig. 2, left and middle) and the vOF from Yeatman's study[31] (right), we examined results from both of these studies (Supplementary Fig. 2). Importantly, we find that our HCP-based VWFA tracts extend more dorsally and anteriorly along the IPS than the vOF[31] consistent with TP-SPL tracts linking inferior temporal and superior parietal cortices[28].

**VWFA structural connectivity with attention network.** The next goal of the analysis was to test predictions of the language node and multiplex models of VWFA function by examining the strength of structural connectivity between VWFA and nodes of the language and attention networks. To enable direct comparison with previous studies of VWFA circuitry, we examined connectivity between VWFA and a priori defined regions of the language and attention networks. Our analysis first focused on examining VWFA connectivity to nodes of the language network associated with reading function[27], including IFG, SMG, AG, and

ITG. Results from the one-sample $t$-test revealed significant connectivity between VWFA and the ITG ($p < 0.001$; Fig. 2b), with an effect size of 0.33 (Supplementary Table 3). In contrast, VWFA connectivity to IFG, SMG, and AG failed to reach statistical criteria. Results were similar using a more anterior SMG region (Supplementary Table 7).

To further probe VWFA connectivity to nodes of the language network, we examined connectivity between VWFA and regions of the STS identified from meta-analysis of 427 fMRI studies examining reading function[33]. Results from the one-sample $t$-tests showed significant VWFA connectivity with anterior and middle STS (Fig. 2c; effect size range: 0.27–0.73; $p < 0.001$). Among subregions of the STS, greatest VWFA connectivity was evident in the mSTS. Additionally, we examined structural connectivity between VWFA and regions of anterior temporal lobe (ATL) based on previous findings which identified a prominent role for the ATL in verbal semantics[34–36]. Results from the one-sample $t$-test indicate strong structural connectivity between VWFA and all ATL subregions ($p < 0.001$; Supplementary Table 6). Together, results show modest to moderate structural connectivity strength of VWFA with ITG, STS/STG, and ATL nodes of the language network. Finally, we examined VWFA connectivity to additional nodes in the language network identified in a previous study of VWFA connectivity[13] including Broca's area, Wernicke's area, and left-hemisphere postcentral gyrus, and results showed significant structural connectivity with Broca's area (Supplementary Table 5).

We next examined the strength of structural connectivity between VWFA and structures of the fronto-parietal attention network. Results revealed a striking pattern of connectivity between VWFA and attention network nodes. Specifically, connectivity between VWFA and FEF, MT+, aIPS, and pIPS showed statistically significant connectivity (Fig. 2d; one-sample $t$-tests; all $ps < 0.001$), with MT+ and pIPS showing the greatest connectivity with VWFA among nodes of the attention network (effect size range: 0.29–2.54). We further explored structural connectivity between VWFA and IPS by exploring connectivity strength in a priori defined retinotopic regions of IPS. Results from the one-sample $t$-test revealed statistically significant

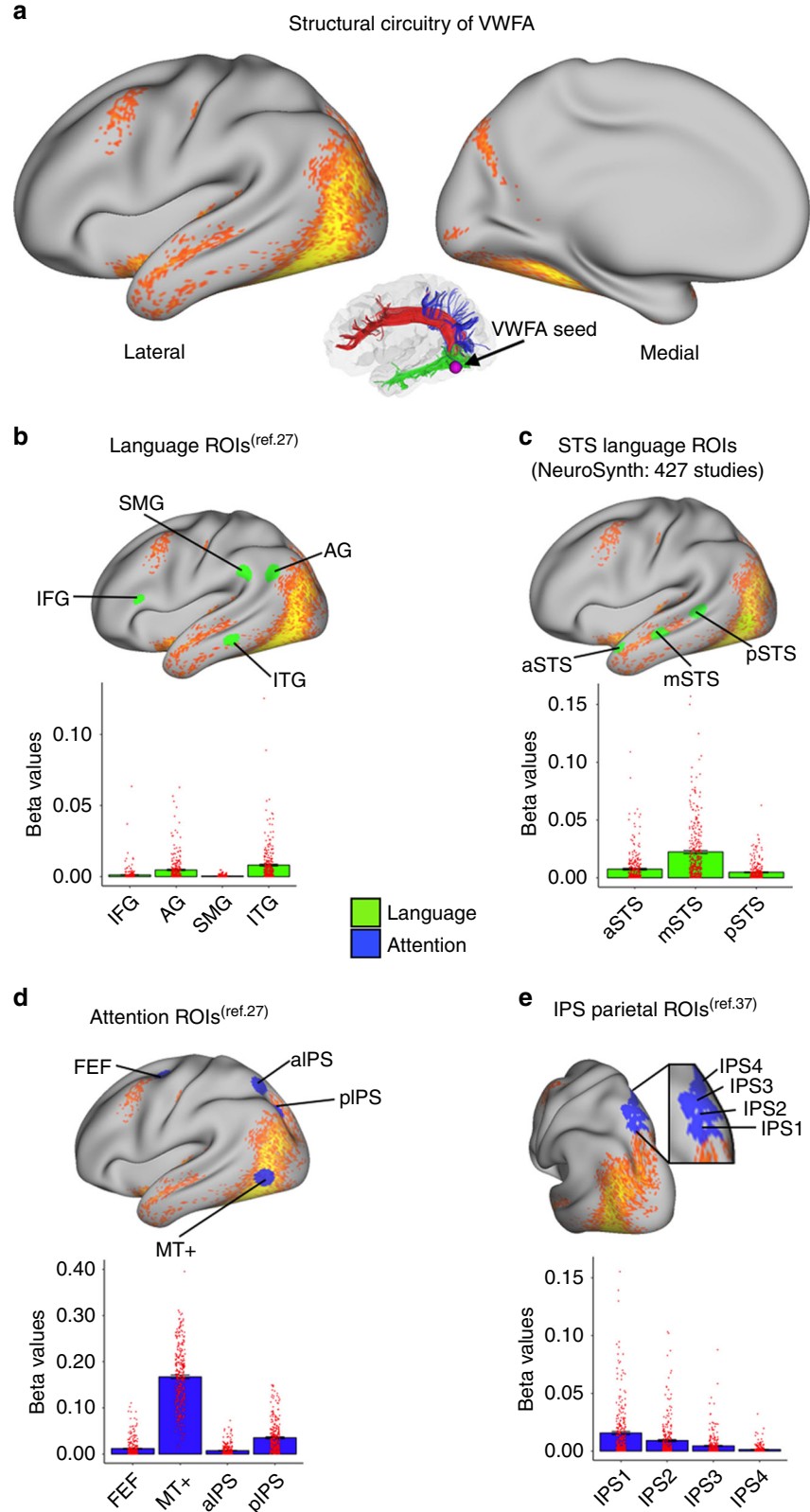

**a** Structural circuitry of VWFA

**b** Language ROIs[(ref.27)]

**c** STS language ROIs (NeuroSynth: 427 studies)

**d** Attention ROIs[(ref.27)]

**e** IPS parietal ROIs[(ref.37)]

connectivity primarily in posterior aspects of the IPS (Fig. 2e; $p < 0.001$) with modest effect sizes ranging from 0.33 to 0.52 (Supplementary Table 3). A notable pattern of connectivity was evident in these retinotopic regions of IPS: structural connectivity of VWFA was greatest in ventral and posterior subregion (i.e., IPS1), with diminishing connectivity in more dorsal and anterior retinotopic subregions (i.e., from IPS2-4).

An additional aim of this analysis was to examine differences in the strength of structural connectivity between VWFA and language network nodes (i.e., from Vogel et al.[27] and the "reading map" of NeuroSynth) relative to connectivity strength between VWFA and fronto-parietal attention network nodes (i.e., from Vogel et al.[27] and the retinotopic map ROIs). Results from the paired-sample $t$-test showed that structural connectivity between

**Fig. 2 Structural connectivity patterns of VWFA. a** Thresholded map showing probabilistic tractography of VWFA seed ($p < 10^{-15}$) in the whole sample ($N = 313$). The location of VWFA is shown as a purple sphere on a map of the white-matter tracts originating from VWFA for a representational subject. All tracts terminate ventrally in the VWFA. **b** Averaged beta values within targeted ROIs in the language network (IFG, inferior frontal gyrus; SMG, supramarginal gyrus; AG, angular gyrus; ITG, inferior temporal gyrus) selected from Vogel et al.[27]. **c** Averaged beta values within anterior, middle, and posterior sections of superior temporal sulcus (STS) selected based on 427 studies from NeuroSynth with the key word "reading"; **d** Attention-related networks (FEF, frontal eye field; aIPS and pIPS, anterior and posterior inferior parietal sulcus; MT+, V5/middle temporal visual area) selected from Vogel et al.[27]. **e** Averaged beta values within the retinotopic ROIs within IPS1-4 identified in Swisher et al.[37]. Error bars in the graph represent the standard error of means. Red dots on each bar represent individual data points.

VWFA and the fronto-parietal attention network overall was significantly greater than connectivity with the language network ($t(312) = -30.54$, $p < 0.001$).

**Whole-brain functional connectivity of VWFA**. The next goal of the analysis was to perform whole-brain functional connectivity analysis of the VWFA. Results revealed VWFA functional connectivity across a distributed network of brain regions in frontal, temporal, parietal, and occipital cortices (Fig. 3a).

**VWFA functional connectivity with attention network**. We next tested predictions of the language and multiplex models of VWFA function by examining the strength of functional connectivity between VWFA and nodes of the language and attention networks. First, functional connectivity between VWFA and nodes of the language network revealed moderate correlations between these regions. Specifically, IFG, SMG, AG, and ITG all showed significant connectivity with VWFA (one-sample $t$-test; $p < 0.001$; Fig. 3b; effect size range: 0.55–0.95; Supplementary Table 4) and relatively weak connectivity strength of VWFA with ITG. To further probe the functional connectivity of VWFA to nodes of the language network, we examined connectivity within regions of the STS identified from meta-analysis of reading function[33]. Results from this analysis showed moderate and statistically significant connectivity between VWFA and all three subregions of the STS (Fig. 3c; effect size range: 0.71–1.00). Among subregions of the STS, greatest functional connectivity was evident between VWFA and pSTS. Strong functional connectivity was also observed between VWFA and ATL regions (Supplementary Table 6). Taken together, similar to results from structural connectivity analysis, functional connectivity results show modest to moderate connectivity strength of VWFA with frontal, temporoparietal and STS/STG nodes of the language network.

We next examined the strength of functional connectivity between VWFA and nodes of the fronto-parietal attention network. Similar to results from structural connectivity analysis, functional connectivity results revealed a striking pattern of connectivity between VWFA and attention network nodes (effect size range: 1.46–1.83; Supplementary Table 4). Specifically, results showed significant functional connectivity between VWFA and all nodes of the fronto-parietal attention network, with particularly strong connectivity in MT+ and IPS (Fig. 3d). To provide additional anatomical detail to our analysis, we explored functional connectivity between VWFA and IPS by exploring connectivity strength in a priori defined retinotopic regions of IPS[37]. Results from this analysis revealed statistically significant connectivity in all subregions (Fig. 3e; one-sample $t$-test; $ps < 0.001$; effect size range: 1.20–1.67; Supplementary Table 4). Furthermore, a similar pattern of functional connectivity was evident in retinotopic regions of IPS as that described in structural connectivity results: VWFA connectivity was greatest in IPS1 subregion, with slightly reduced connectivity in IPS2-4.

A final aim of this analysis was to examine differences in the strength of functional connectivity between VWFA and language network nodes relative to connectivity strength between VWFA and fronto-parietal attention network nodes. Results from the paired-sample $t$-test showed that functional connectivity between VWFA and the fronto-parietal attention network was significantly greater than connectivity with the language network ($t(312) = -25.03$, $p < 0.001$; Supplementary Table 4).

**VWFA circuits predict attention and language abilities**. The next major goal of the study was to test predictions of the language node and multiplex models of VWFA function and examine the relationship between VWFA circuitry and individual differences in language and visuo-spatial attention abilities.

We first examined brain-behavior relationships in the context of structural connectivity and performed multiple regression analyses in which independent variables were either VWFA connectivity to language or attention node ROIs while dependent variables were standard scores of language or visual attention abilities. In the statistical analysis, we tested the difference between structural connectivity regression models and a baseline model. Results from the multiple regression analysis showed a striking pattern of double dissociation in which structural connectivity between VWFA and nodes of the language network, instantiated in the STS, uniquely predicted both picture vocabulary ($\Delta R^2 = 0.031$, $p = 0.007$; FDR corrected $p = 0.014$; BF = 20.859; Supplementary Table 8) and word reading abilities ($\Delta R^2 = 0.025$, $p = 0.025$; FDR corrected $p = 0.25$; BF = 0.721; Supplementary Table 9) while connectivity between VWFA and nodes of the fronto-parietal attention network uniquely predicted visuo-spatial attention abilities ($\Delta R^2 = 0.054$, $p = 0.004$; FDR corrected $p = 0.012$; BF = 5.731; Supplementary Table 10) in addition to the effect of gender. Consistent with double dissociation, structural connectivity between VWFA and nodes of the language network failed to predict age-adjusted scores of attention (multiple regression; $\Delta R^2 = 0.012$, $p = 0.964$), and connectivity between VWFA and nodes of the attention network failed to predict language abilities (multiple regression; $\Delta R^2 = 0.003$ for picture vocabulary, $\Delta R^2 = 0.009$ for word reading; $p > 0.05$; Fig. 4). A nonparametric regression model revealed similar results, and provided only a subtle advantage relative to parametric results, with the proportion of variance explained increasing by 1.7–6.8% (Supplementary Table 14). Additionally, the use of nonparametric Spearman correlations to examine the relationship between predicted and actual scores from VWFA structural connectivity also showed similar results (Supplementary Table 15).

We next performed multiple regression analyses in which independent variables were either VWFA functional connectivity to language or attention node ROIs, while dependent variables were age-adjusted scores of language or visuo-spatial attention abilities. Results from these analyses failed to show any significant brain-behavior relationship (multiple regression; $\Delta R^2 \leq 0.011$ for all analyses; Supplementary Tables S8–10). Finally, we examined whether a full regression model, which included both structural

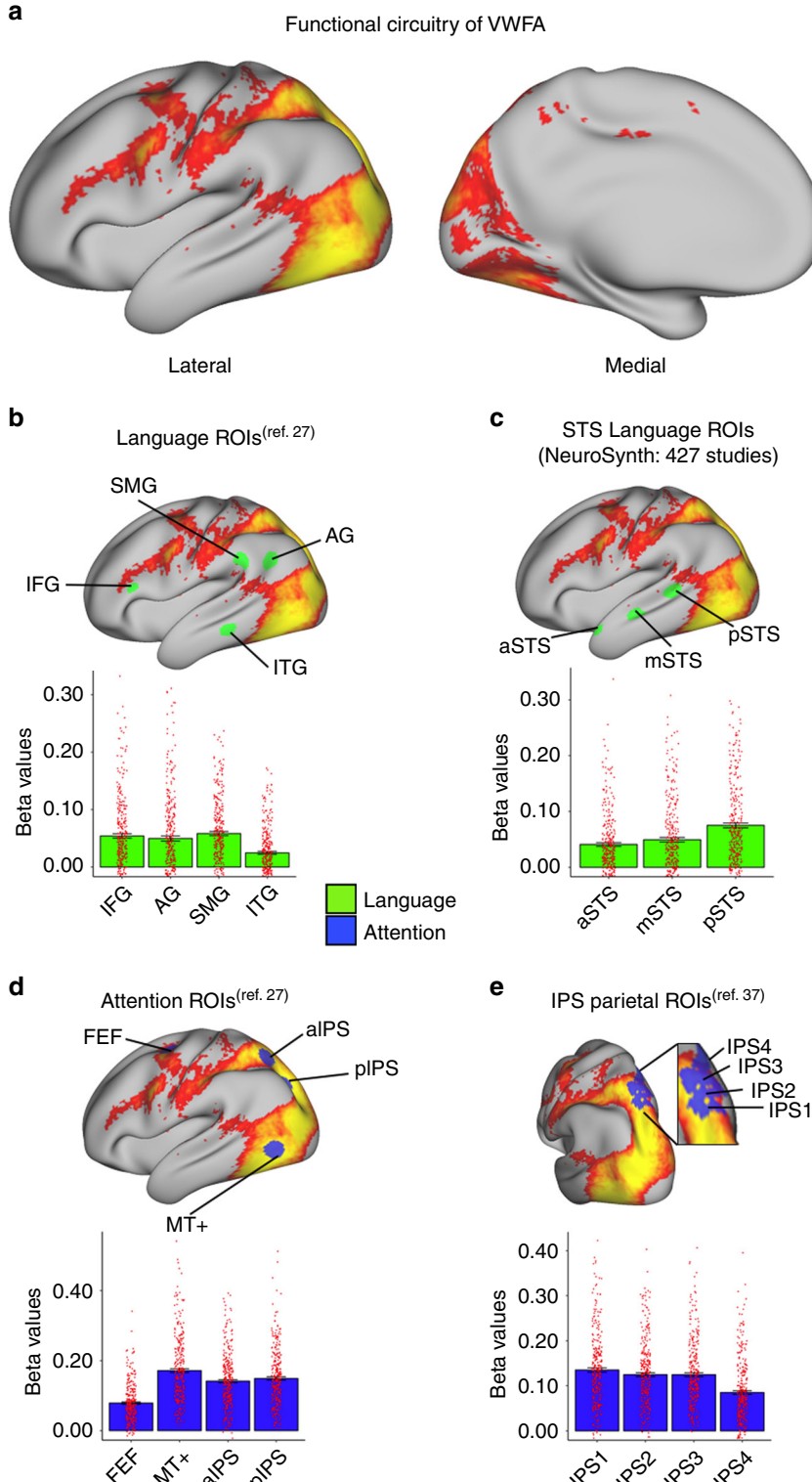

**Fig. 3 Intrinsic functional connectivity of VWFA. a** Thresholded map showing intrinsic functional connectivity of VWFA seed ($p < 10^{-40}$) in the whole sample ($N = 313$). **b** Averaged beta values within targeted ROIs in the language network (IFG, SMG, AG, and ITG) selected from Vogel et al.[27]. **c** Averaged beta values within anterior, middle, and posterior sections of STS selected based on 427 studies from NeuroSynth with the key word "reading". **d** Attention-related networks (FEF, aIPS, pIPS, and MT+) selected from Vogel et al.[27]. **e** Averaged beta values within the retinotopic ROIs within IPS (IPS1-4) identified in Swisher et al.[37]. Error bars in the graph represent the standard error of means. Red dots on each bar represent individual data points.

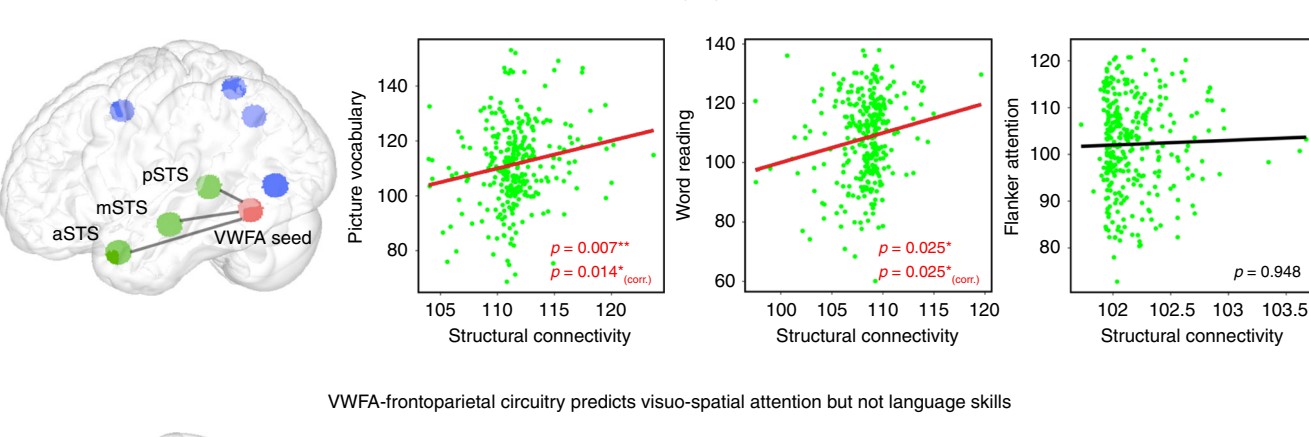

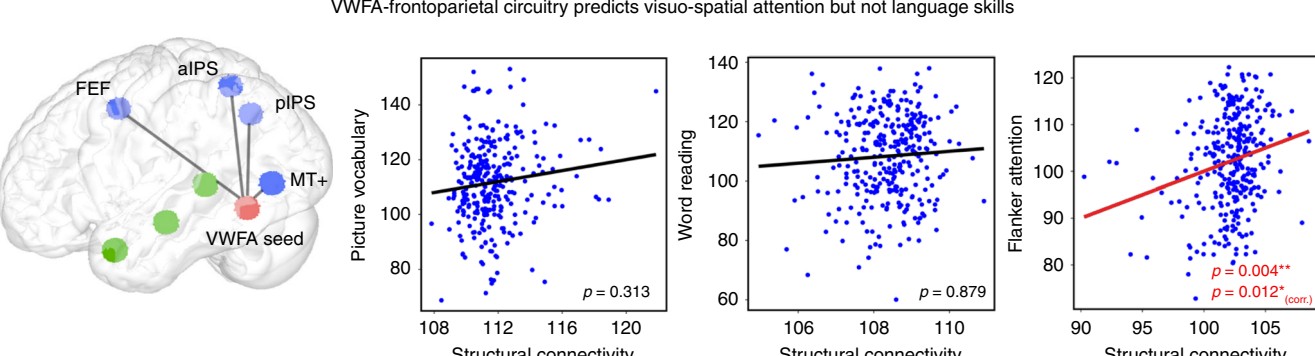

**Fig. 4 VWFA circuits differentially predict language and visuo-spatial attention abilities across individuals. a** Structural connectivity of VWFA with STS nodes (anterior, middle, and posterior STS) was significantly correlated with individuals' performance on picture vocabulary and word reading tasks, but not on the Flanker visuo-spatial attention task. **b** Structural connectivity of VWFA with fronto-parietal attention network nodes (FEF, aIPS, pIPS, and MT+) was significantly correlated with individuals' performance on the Flanker attention task but not on the word reading or picture vocabulary tasks. Y-axis is age-adjusted performance scores on cognitive tasks and x-axis presents the predicted performance scores on cognitive tasks from the probability of structural connectivity of VWFA to either the language or the attention network ROIs.

and functional connectivity measures, better predicted language and attentional standard scores compared to models that included either structural or functional connectivity measures alone. In all cases, the structural connectivity model was the best predictor of language and attentional abilities (Supplementary Tables S8–10). Finally, brain-behavior analyses were also performed using ATL subregions, and results revealed that functional connectivity between VWFA and ATL predicted picture vocabulary scores (multiple regression; $\Delta R^2 = 0.027$, $p < 0.01$; BF = 2.896; Supplementary Table 11). The modest $R^2$ values from our regression analyses may reflect our strong theoretically-driven approach for ROI selection; future studies employing a data-driven approach may facilitate the identification of additional sources of behavioral variance.

**Task-related functional modulation of VWFA connectivity.** Finally, given inherent issues with interpretation of functional connectivity from rs-fMRI alone, we examined task-based engagement and connectivity of VWFA circuits during reading and attention tasks. Briefly, analysis of activation during both reading and Flanker attention tasks revealed that a broad extent of posterior vOTC, including the VWFA ROI used in structural and intrinsic functional connectivity analyses, is engaged during both tasks (Supplementary Fig. 3). Moreover, the reading task elicited increased connectivity between the VWFA and nodes of both the language and attention networks (Supplementary Table 12), and the VWFA showed increased connectivity with all nodes of the attention network during the Flanker task (Supplementary Table 13; see Supplementary Discussion). These results demonstrate that task-based functional connectivity results show

strong convergence with findings from structural and intrinsic functional connectivity, and highlight a tight link between intrinsic VWFA circuits and cortical function.

## Discussion
VWFA is a critical brain region for reading and is widely considered a key node of the brain's language network, however detailed whole-brain analysis of VWFA structural and functional circuitry using a large and high-quality dataset has been absent from the literature, precluding a more comprehensive understanding of its large-scale architecture. Converging results from probabilistic tractography and seed-based functional connectivity revealed that VWFA has moderate-to-strong connectivity with nodes of the brain's language network, particularly STS regions associated with phonological processing, as well as fronto-parietal attentional systems, particularly the IPS. The strength of structural and functional connectivity between VWFA and nodes of the attentional system was greater than those measured between the VWFA and the language network. Crucially, brain-behavior analyses revealed a double dissociation: the strength of structural connectivity between VWFA and nodes of the language network predicted individual differences in reading and language abilities but not attention, whereas structural connectivity between VWFA and nodes of the dorsal fronto-parietal attentional network predicted visual attention abilities but not reading and language abilities. Finally, activation and functional connectivity during reading and attention tasks showed strong convergence with structural and intrinsic functional connectivity profiles. Taken together, results support a multiplex model of VWFA function in which VWFA has a more general role in visual function beyond

reading and is tightly linked to fronto-parietal attention systems. We propose that the VWFA is a gateway linking the language and attention system which functions by shining an attentional "spotlight" on visual representations in the vOTC as a means of amplifying these representations for use by distinct functional systems.

A role for the VWFA in reading function has been a foundation of cognitive neuroscience for decades[2,5,14,17], and more recent work has highlighted the structural and functional circuitry linking VWFA to regions associated with language function as a crucial aspect of the brain's reading network. For example, results from a structural connectivity study showed that more anterior aspects of the VWFA showed strong connectivity with a wide extent of temporal cortex including anterior, mid, and posterior STG/STS[10] and the ATL[34–36]. A second structural connectivity study identified the inferior longitudinal fasciculus as the key white matter tract connecting VWFA to superior temporal cortex[11]. Furthermore, recent studies showed enhanced functional connectivity profiles between individually localized VWFA seeds and posterior superior temporal cortex, and that the strength of this particular link was related to performance on a semantic classification task involving written words[12,13].

Results from the current study build on previous findings and are the first to reveal the detailed structural and functional architecture of the VWFA in the same group of participants and provide new information regarding VWFA's links to the language network. First, results show distinct patterns of connectivity between VWFA and several established nodes of the brain's language network. Specifically, converging results from both probabilistic tractography and seed-based functional connectivity reveal stable patterns of connectivity between VWFA and nodes of the language network, with regions of the STS showing prominent patterns of structural and functional connectivity, but with weaker connectivity patterns for lateral prefrontal (IFG or Broca's area) and parietal (SMG and AG) brain systems canonically associated with reading and language functions. These results highlight the centrality of temporal lobe structures to the language network[38], and support the use of well-powered meta-analyses, such as Neurosynth[33], for deriving ROIs for functional brain circuit analysis. Furthermore, results from brain-behavior analyses provide additional new information regarding VWFA circuits underlying critical aspects of language functions. Specifically, results revealed that the strength of structural, but not functional, connections linking VWFA to STS subregions of the language network predict individual differences in picture vocabulary and word reading abilities.

An important theoretical question is what functional role does the VWFA to STS circuit play during reading? STS is a brain region that has been consistently implicated in phonological processing[39–41], and the VWFA to STS link may play a key role in mapping orthographic representations to phonologic representations during reading. An interesting strand of evidence supporting this hypothesis comes from an intrinsic functional study of the congenitally deaf which revealed a specific reduction in VWFA to STS connectivity in deaf individuals while all VWFA connections to lateral prefrontal and inferior parietal language regions were conserved[42]. This result suggests that the presence of robust phonological representations in STS, which are thought to be greatly diminished in deaf individuals, may facilitate the strengthening of functional and structural connections between VWFA and STS during development. The importance of this particular link may also be considered in the context of reading acquisition. When children first learn to read, they initially rely heavily on "sounding out" words that they are reading, a process which may significantly rely on and strengthen anatomical and structural connections between VWFA and STS. Consistent with

this view, one recent study showed that reduced structural connectivity of VWFA with multiple temporal regions, including STS, distinguished children with word reading difficulties and typically-developing children[43]. Together, results from multiple lines of evidence highlight a critical role for the VWFA-STS circuit in reading function.

While most previous studies investigating the structural and functional circuitry of VWFA have focused on patterns of connectivity with nodes of the language network[12,13], recent evidence supports the hypothesis that VWFA is also tightly linked to the fronto-parietal attention network. For example, both structural and functional connectivity studies[10,26,27] have identified prominent links between VWFA and fronto-parietal regions involved in attention processing, including MT+, IPS, and FEF. However, a primary focus on language systems in the VWFA literature has diminished consideration of these fronto-parietal attention regions. One previous rsFC study particularly highlighted intrinsic functional connectivity between VWFA and fronto-parietal regions involved in attention processing[27]. This study showed that the VWFA has strong intrinsic functional connectivity with fronto-parietal regions involved in attention processing, and that the strength of these connections exceeded intrinsic connectivity with nodes of the language network. There were two limitations of this work that precluded a more complete understanding of the functional significance of these VWFA connections. First, while this study reported a link between VWFA-IPS connectivity and reading ability, this brain-behavior effect was based on a combined sampling of children and adult participants who showed marked differences in reading ability. Moreover, this study did not include a behavioral task explicitly probing visual attention abilities, which is important for a more thorough understanding of the functional role for VWFA-IPS connectivity.

Results from the current study provide new information regarding structural and functional connectivity of the VWFA to the fronto-parietal attention network. First, converging evidence from both probabilistic tractography and seed-based functional connectivity reveal robust patterns of connectivity between VWFA and nodes of the fronto-parietal attention network. Consistent with previous work[27], the strength of VWFA connectivity to these regions was greater than connectivity with nodes of the language network. Critically, results from the current study showed that the strength of structural connectivity between VWFA and the fronto-parietal attention network predicts visual attention abilities. Additionally, our study provides new details regarding anatomical and functional specificity of VWFA connectivity to specific retinotopic subregions of the IPS[37]. Results show the greatest connectivity between VWFA and the posterior IPS, particularly IPS1 which has stronger connectivity with visual cortex than more anterior IPS regions[44]. The specific functions and connectivity of subdivisions of IPS in both human and other primates have been widely studied[45]. Compared to anterior IPS subregions (IPS3/4), posterior aspects of IPS, i.e., IPS1/2, have been suggested to more likely to connect with retinotopically defined visual regions[46], and are more engaged in saccadic eye movements[47], visuospatial working memory[48,49], top-down attention[50], and object-variant representations[45,51,52]. Therefore, the robust connections of VWFA and posterior IPS subregions may help sustain visuo-spatial attention and working memory resources for word reading, probably because processing visual words require granular information of visual features of elements as well as visuo-spatial arrangement of different elements.

Converging results from the current study and previous studies[26,27] strongly suggest a prominent role for attentional systems in the context of VWFA function. The fronto-parietal attentional system is a distributed network associated with the

control of attention[53–55], eye movements[45,56], and visuo-spatial perception[57,58], and these particular functions all represent critical aspects of reading. For example, proficient readers saccade four to five times per second while reading, and regions of the fronto-parietal attentional systems, including FEF and IPS, have long been implicated in this process[56,59]. Moreover, a prominent neural model of attention states that these particular regions are critical for preparing for, and executing, goal-directed visual behaviors[60], such as directing the eyes across a line of script when reading. Importantly, this model further proposes that a critical function of the fronto-parietal attentional system is to amplify visual representations within the attentional window. In the context of reading, we hypothesize that fronto-parietal brain systems guide visual attention as a means of amplifying representations of written words in the VWFA so they may be distributed to nodes of the language network for subsequent phonological and lexical processing. Brain-behavior analyses support this hypothesis by showing that structural connectivity of VWFA to distinct brain systems supports different aspects of visual and reading function: while structural connectivity between VWFA and the fronto-parietal attentional network predicts visual attention, connectivity between VWFA and the language network predicts reading and language function.

The precise functional role of the VWFA has been hotly debated in the cognitive neuroscience literature for many years[1,14,16]. Results from the current study strongly support the multiplex model of VWFA function in several important ways. First, converging results from both probabilistic tractography and functional connectivity show that while VWFA has prominent connections with the language network, this region also has extensive and even stronger connections with the fronto-parietal attention network. Moreover, findings from brain-behavior analysis not only revealed a relationship between language abilities and the strength of structural connectivity between VWFA and STS language nodes, but also identified a relationship between (non-linguistic) visual attention abilities and connectivity between VWFA and the fronto-parietal attention network. Together, results indicate that neither connectivity patterns of VWFA nor its behavioral relations are specific to reading or language function. Rather, results have identified connectivity patterns and behavioral abilities that extend beyond the language domain, encompassing critical brain systems and abilities associated with visual attention. Based on these findings, we suggest that the VWFA serves as a critical gateway in the vOTC that links three distinct brain systems, including low-level visual, attentional, and language processing.

In conclusion, we adopted a network connectivity approach to understand the properties of distributed circuits in a putatively functionally-specialized region. We examined structural and functional connectivity with high-resolution imaging in a large cohort to test competing models of VWFA function with a strong theory-driven approach. Converging results indicate that VWFA is tightly coupled to both language and attention networks in the brain, and that language and attentional abilities are reflected in connectivity patterns between VWFA and their respective networks. Results strongly suggest that VWFA is not specialized for just for reading and language function, but rather plays a broader role in integrating language and attention. Taken together, our study highlights the multiplexed VWFA circuits underlying reading, a highly-demanding visual decoding task that requires rapid online visual perception, attention, eye movement coordination, and conversion to language representations[20]. Finally, our approach is very much in resonance with the broader framework of "connectivity fingerprints"[61] or "connectivity constrained cognition"[62]. Consistent with this framework, we demonstrated that the specific functions of VWFA are very likely to depend on its functional and structural connectivity patterns with other brain networks. This organizational principle may also be useful for elucidating the functions of other brain regions.

## Methods

**Data**. The Washington University Institutional Review Board (IRB # 201204036) approved all procedures for the Human Connectome Project (HCP). Informed consent was obtained from all participants and all open access data were deidentified.

**Participants**. Participants consisted of 313 adults from the Human Connectome Project (HCP) obtained under the HCP 1200 data release (188 females; age range: 26–35 years old; details of demographic information and behavioral data can be found in Supplementary Tables 1 and 2). The participants were selected from 1206 subjects based on the following criteria: (1) age between 26 and 35 years old; (2) right-handed, defined as a handedness score >0; (3) completed all four runs of 3T resting-state fMRI (rs-fMRI) protocol; (4) averaged scan-to-scan head motion for rs-fMRI data is less than 0.2 mm; (5) completed the diffusion MRI protocol (See Fig. 1a for the selection pipeline), yielding a total of 313 participants.

**Data acquisition and preprocessing**. Diffusion MRI data were acquired at Washington University in St Louis using a customized Siemens Magnetom Connectome 3T scanner with a 100 mT/m maximum gradient strength and a 32 channel head coil[63,64]. Using a single-shot, single refocusing spino-echo, echo-planar imaging sequence with 1.25 mm isotropic spatial resolution (TE = 89.5 ms, TR = 5520 ms, flip angle = 78 degree, refocusing flip angle = 160 degree, FOV = 210 × 180 mm), three different gradient tables of 90 diffusion weighting directions with two different were collected over 6 runs and 6 b = 0 images were also acquired in each run. Diffusion weighting consisted of 3 shells of b = 1000, 2000, and 3000 s/mm² and two phase-encoding direction-reversed images composing all runs. Total acquisition time for the 6 runs was approximately 1 hour. Minimally preprocessed Diffusion MRI data were obtained from the HCP[65].

For each individual, 1200 frames were acquired using multiband, gradient-echo planar imaging with the following parameters: RT, 720 ms; echo time, 33.1 ms; flip angle, 52°; field of view, 280 × 180 mm; matrix, 140 × 90; and voxel dimensions, 2 mm isotropic. During scanning, each individual was eye-fixated on a projected crosshair on the screen. A total of 4 15-minute runs of rs-fMRI were acquired in each participant. Minimally preprocessed resting-state fMRI data were obtained from the HCP[65].

**Regions of interest (ROIs)**. Consistent with previous studies[27], the VWFA was defined as a 6 mm radius geodesic disc centered on the MNI coordinate (−45, −57, −12). This seed location was used to facilitate comparisons to previous studies of VWFA connectivity. For all the coordinate-based ROIs, a disk with a diameter of 6 mm on the cortical surface map was transformed from volumetric space to surface space using the connectome workbench (https://www.humanconnectome.org/software/get-connectome-workbench).

Language network ROIs were constructed using coordinates previously published in a whole-brain study of VWFA connectivity[27]. That study identified ROIs for left-hemisphere pars triangularis (BA 45) of inferior frontal gyrus (IFG), angular gyrus (AG), supramarginal gyrus (SMG), and inferior temporal gyrus (ITG). ROIs for these regions were constructed as 6 mm radius geodesic discs centered on the following MNI coordinates: IFG, (−53, 27, 16); AG, (−49, −57, 28), SMG, (−56, −43, 31), and ITG, (−61,−33,−15). To further explore VWFA connectivity to nodes of the brain's language system, we constructed additional ROIs in superior temporal sulcus (STS), a key region for phonological and multimodal language processing[41,66]. Coordinates for STS ROIs were identified using the association test activation map in Neurosynth using the search term "reading," which yields a statistical inference map based on 427 studies[33]. Peaks in this map were identified in anterior (aSTS; −54, 9, −20), middle (mSTS; −53, −18, −10), and posterior STS (pSTS; −52, −40, 5). Additionally, to enable comparisons with a recent study investigating VWFA connectivity with the language network[13], additional nodes of the language network were defined as pars opercularis (BA 44) of the inferior frontal gyrus (IFG), the parietal operculum of inferior parietal lobule, and precentral gyrus (preCG). These regions were chosen from peak activations for viewing words compared to pictures of objects in a previous study[13], and are associated with putative Broca's area, Wernicke's area, and speech motor cortex. ROIs for these regions were constructed as 6 mm radius geodesic discs centered at MNI coordinates (−45, 14, 21) for the pars opercularis, (−49, −40, 26) for parietal operculum, and (−57, −11, 45) for preCG. The location of these ROIs are also similar to a second study examining connectivity of VWFA to the language network[12]. Additionally, the Neurosynth "reading" map also identified an additional a peak in the SMG, located at (−44, −38, 40), anterior to the SMG ROI used by Vogel et al.[27].

Finally, the anterior temporal lobe (ATL) has been consistently implicated in semantic aspects of reading and language[67]. We, therefore, included four ATL ROIs from previous studies[34–36] to examine their connectivity with VWFA. Coordinates for all ATL regions are listed in Supplementary Table 6. Note that the

HCP project employs a single echo protocol, which could potentially lead to signal loss in ATL regions sensitive to susceptibility artifacts[68]. However, even with this limitation, we successfully detected significant intrinsic connectivity of VWFA to multiple ATL regions.

Based on results from a previous study showing intrinsic functional connectivity between VWFA and fronto-parietal attention networks[27], 6 mm geodesic discs were constructed for four ROIs, including left-hemisphere frontal eye fields (FEF; −26, −5, 50), left V5/middle temporal visual area (MT+; −45, −71, −1), and left anterior (aIPS; −25, −62, 51) and posterior intraparietal sulcus (pIPS; −25, −69, 34). To provide additional anatomical detail regarding structural and functional connectivity between VWFA and subregions of the IPS, four additional ROIs were constructed in the IPS based on retinotopic maps in IPS identified in a previous study[37].

**Data analysis.** Our data analysis and analytical procedures are illustrated in Fig. 1b. Structural connectivity for each voxel in the whole-brain was computed in each individual space from diffusion MRI data following the same pipeline as in Lascano et al.[69]. For each surface vertex in the grey-white matter interface, the cortico-cortical connectivity through probabilistic tractography was computed using a constrained spherical deconvolution model of within-voxel diffusivity[70]. These techniques have been shown to improve the ability to track crossing and fanning fibers, however forking fibers continue to represent a methodological challenge using current DTI techniques. We also employed well-established method[71] to correct for tract distance. The FreeSurfer-obtained white matter surface was shrunk 3 mm into the white matter to avoid superficial tracts[72] and was then seeded 5000 streamlines per vertex. The proportion of streamlines connecting two vertices was interpreted as the probability that a white matter axonal bundle connects both cortical points, namely tract strength[73]. The decision to use 5000 streamlines as the number of trials was based on pilot studies and the observation that tract strengths stabilizes at this trial number across a holdout sample of vertices and participants. To assess population-wise connectivity of the VWFA, we first calculated the probability of all cortico-cortical connections at the vertex level for each subject, estimated the 95-percentile value of the probability of all connections for the subject, and computed the median value across participants (0.00402). We then used this population mean as the empirical null value for our statistical analysis, and tested the hypothesis whether vertex-based connectivity is significantly greater than this threshold. The rationale for controlling for age and gender in both structural and functional analyses is that both of these factors have been shown to influence connectivity patterns in large-scale brain network analyses[74,75] and the resulting connectivity map was then thresholded at $p < 10^{-15}$ (FDR corrected) to illustrate patterns of structural connectivity.

Surface-based resting-state functional connectivity (rsFC) for each grayordinate vertex was computed from the FIX-denoised grayordinate-space rs-fMRI data in each subject. For each cortical vertex, the correlation between its time course and the average VWFA time course was computed to produce an $r$-map, where the $r$ value of each vertex represents the extent of its time-series correlation with VWFA. The $r$-map was then transformed into a $z$-map by applying the Fisher $r$-to-$z$ transformation. A one-sample $t$-test was then computed at each vertex to examine whether the group-mean $z$-value across subjects was significantly different from zero with FDR correction, while controlling for age and gender. The group-averaged $t$-map and $p$-map were generated, and thresholded at $p < 10^{-40}$ (FDR corrected).

Connectivity between VWFA and the language and attention networks was performed using a theoretically-driven, ROI-based approach. We first computed connectivity betas for each link between VWFA and each node of the language network, using a priori ROIs from Vogel et al.[27] and the Neurosynth meta-analysis map for the search term "reading". We then computed the average beta value across these links for each participant. This procedure was then performed for the links connecting VWFA to the visuospatial attention network using a priori ROIs from Vogel et al.[27] and retinotopic maps of IPS. These betas, which represent the average connectivity between VWFA and the language and attention networks for each participant, were then compared using a paired-samples $t$-test. This analysis was conducted separately for intrinsic functional and structural connectivity measures.

**Behavioral measures.** To examine the relationship between VWFA connectivity and language and attention abilities, we used a total of three behavioral measures, including two languages (word reading and picture vocabulary) and one attention measure (Flanker). Two language tasks, collected as part of the HCP protocol, were used to examine the relationship between VWFA connectivity and language abilities. The Oral Reading Recognition Test from the NIH Toolbox (TORRT) measures single letter and word reading, and indexes exposure to written text and decoding skills, including phonological processing[29]. The Toolbox Picture Vocabulary Test (TPVT) assesses vocabulary size and spoken word comprehension. In this task, participants were presented with four pictures and a spoken word and participants were asked to select the picture matching the spoken word[30].

To examine the relationship between VWFA connectivity and visual attention abilities, the Eriksen Flanker task (Flanker) was used. This measure assesses participants' visual attentional skills and inhibitory control[76]. In this task, participants indicate the left-right orientation of a centrally presented stimulus in the context of a string of similar stimuli (e.g., left and right arrows). In some trials, the center stimulus has the same orientation as the surrounding ones (e.g., left arrow in the middle of 4 left arrows; congruent trials), but in other trials the center stimulus has a different orientation with the surrounding ones (e.g., left arrow in the middle of 4 right arrows; incongruent trials). Scores were calculated based on accuracy and reaction times.

Age-adjusted standardized scores for the TORRT, PVTT, and Flanker tasks were used in brain-behavior analyses. Age bands were used (18–29, 30–39, 40–49, 50–59, 60–69, 70–79, 80–85) to calculate means and standard deviations for each age group, and standardized score for each subject were generated based on the mean and standard deviation values of the corresponding age group that the subject belongs to (for details, see Supplementary Methods).

**Brain-behavior analyses.** Multiple regression analyses were performed to examine the relationship between structural and functional connectivity measures and behavioral measures of language and attention abilities. These analyses examined whether structural and functional connectivity between VWFA and target ROIs in language and dorsal attention networks could predict individual differences in cognitive task performance when controlling for participants' gender. Age was controlled for in these analyses by using age-adjusted standardized scores from the behavioral tasks as dependent variables in the regression models. Regression analyses separately examined effects of functional and structural connectivity, and therefore four models were constructed in hierarchy: (i) a baseline model in which only gender was included; (ii) a functional connectivity (FC) model in which gender and the averaged rsFC beta values between VWFA and target ROIs, instantiated in either language (see Fig. 4, top) or attention network nodes (see Fig. 4, bottom), were included; (iii) a structural connectivity (SC) model in which gender and the averaged SC beta values between VWFA and target ROIs, instantiated in either language or attention network nodes, were included; and, (iv) the full model in which gender and the averaged rsFC and SC beta values between VWFA and target ROIs, instantiated in either language or attention network nodes, were included. ANOVAs were used to test the difference of FC and SC models over the baseline model, and the full model over the FC and SC models. Cohen's $f^2$[77] and Bayes Factor[78] were calculated for the regression models to reveal the effect size and the evidence strength of each model. Planned tests included VWFA-language connectivity predicting two language measures (word reading and picture vocabulary) and VWFA-attention connectivity predicting the attention measure (Flanker task). Given our a priori hypotheses regarding these aspects of VWFA connectivity, these analyses were performed with FDR correction at the $p < 0.05$ level (note that we report FDR corrected and uncorrected p-values for all tests). A second component of the brain-behavior analyses examined the specificity of these initial finding. Specifically, these latter analyses examined whether: (a) connectivity between VWFA and language nodes predicted Flanker task scores, and (b) connectivity between VWFA and attention nodes predicted language scores. We did not require corrections for multiple comparisons for this second aspect of the brain-behavior analysis to enable a more stringent test of specificity for the initial set of results. Finally, in addition to the parametric linear regression analysis, we also performed a general additive model (GAM) analysis, which incorporates cubic regression spline[79] to examine brain-behavior relationships. We compared the model fit of the parametric linear model with the GAM on the same measures, and used Spearman correlations to examine the relationship between the predicted values and the actual values on each behavioral measure.

**Task fMRI data for word reading and visual attention.** To investigate engagement and functional connectivity of the VWFA during reading and attention tasks, and address the final goal of this study, we analyzed two task fMRI datasets including word reading/rythming (https://openneuro.org/datasets/ds000003/versions/00001) and visuospatial attention/Flanker tasks (https://openneuro.org/datasets/ds000102/versions/00001) from OpenNeuro. We first examined task activation in both tasks within the VWFA, and then we used the generalized psychophysiological interaction (gPPI) model[80] to examine task-based functional connectivity, and employed the same VWFA seed and language and attention network targets described in the structural and intrinsic functional connectivity analyses from the HCP dataset for this analysis (details in the Supplementary Methods).

**Reporting summary.** Further information on research design is available in the Nature Research Reporting Summary linked to this article.

## Data availability
The data are available through the Human Connectome Project (https://www.humanconnectome.org/) and the two task fMRI data are available through the OpenNeuro website (https://openneuro.org/). The codes for processing the diffusion MRI data are available from https://zenodo.org/record/3517201#.XbRnwCWxWEc.

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

## Acknowledgements

This work was supported by the National Institutes of Health (K01MH102428 to D.A.A, and HD094623, HD059205, MH084164 to V.M.), the Brain & Behavior Research Foundation (NARSAD Young Investigator Grant to D.A.A) the European Research Council (ERC Starting Grant Agreement No. 757672: NeuroLang to D.W.), the Institut National de Recherche en Informatique et en Automatique (INRIA associated team LargeBrainNets), and the France-Stanford Center Visiting Junior Scholar Fellowship to L.C. Data were provided by the Human Connectome Project, WU-Minn Consortium (Principal Investigators: David Van Essen and Kamil Ugurbil; 1U54MH091657) funded by the 16 NIH Institutes and Centers that support the NIH Blueprint for Neuroscience Research; and by the McDonnell Center for Systems Neuroscience at Washington University.

## Author contributions

L. Chen, D. Wassermann, D. Abrams, and V. Menon designed the research; L. Chen, D. Wassermann, J. Kochalka, and G. Gallardo-Diez analyzed the data; L. Chen, D. Abrams, D.Wassermann, and V. Menon wrote the paper; and all authors contributed to editing it.

## Competing interests

The authors declare no competing interests.
