## [Peer Review File · Nature Communications]

Reviewers' comments:

Reviewer #1 (Remarks to the Author):

This is an ambitious study to evaluate the connectivity of the VWFA using structural tractography and rs-fMRI. It clearly represents a significant amount of work.

Specific comments:

p.4: classically, the function of this vOT region was influenced by neuropsychological data (from Dejerine onwards). The Behrmann and Plaut studies are important. In addition, the largest cases series of such patients (Roberts et al., *Cerebral Cortex* 2012) is not considered at all. That study shows that there are co-occurring non-reading deficits across the patients including weakness with visually-complex novel and familiar stimuli including faces. Given the breadth of assessment and number of patients reported, this study and its findings need to be considered in the paper. This study sets out many reasons why reading is a highly-demanding visual decoding task (which requires rapid online visual perception, attention, eye movement programming – see Leff et al, as well as conversion to language representations).

p.4: vertical occipital fasciculus – it is claimed here that this fasciculus connects VWFA to IPS. Is this correct? My understanding was that this fibre bundle is much more posterior. Even in the authors' own Figure 2, the fibres terminate posterior to the VWFA marked seed and project dorsally to the occipito-parietal junction (not to IPS).

There is a more fundamental assumption that should be addressed. The study examines white-matter connectivity. It is likely that these structural pathways are present long before reading tuition commences and thus this neural network must reflect more basic primary visual and language functions – which reading is parasitic on. Although there is evidence that experience can modulate strength of connection and cortical volume, as far as I am aware there is no evidence that experience can generate new patterns of connectivity across the cortex.

Methods

p.7: although the HCP undertook distortion correction methods for the DWI, the rs-fMRI were collected with a single echo and various aspects of the anterior temporal lobe are known to be missing. This will limit the range of language-semantic related regions that can be examined via the rs-fMRI data. These regions are known to be involved in reading (see Hoffman et al., *PNAS* 2015; Woollams et al *PNAS* 2017).

p.8: it would be useful to include anterior temporal regions in the language-related ROIs (see point above). Which is reinforced given the primary connection along the ILF.

p.10: tract strength was interpreted as the proportion of streamline between the VWFA and each ROI. Note that streamline numbers are influenced by many factors other than point-to-point connectivity including forking, crossing and fanning fibres. The values are also influenced by tract distance. How were these confounds dealt with?

p.10: a t-test against zero was used to assess the presence of a connection. I do not understand this. Surely all probable values (proportion of streamlines) will be >0 by definition. No negative values will be present. Isn't a different baseline needed? E.g., the level of connectivity between randomly-selected ROIs? There will be a minimal number of streamlines simply by chance.

p.11-12: there were multiple behaviour measures for language and attention. Which of these were used in the brain-behaviour regression analysis and how? Were the analyses repeated for each measure or where the measures combined in some way?

p.13-14: as noted above – I do not understand how "significant" connections to the ROIs is being

assessed or what the "effect size" is in this context?

Apologies if I have missed this in the Methods section but I do not understand how the comparison of connectivity (structural or functional) between VWFA and either the language or attention networks is computed? Are the values for the different connections across people combined in some way?

Apologies again – but what method was used to extract the ILF, AF and vOF as shown in Figure 2?

Discussion

I note that the R^2 for the regression models are not very high. Thus it might be important not to overstate the brain-to-behaviour links. The connectivity-behavioural correlations show some limited relationship but the vast majority of the behavioural variance is left unexplained.

It is probably also important to be careful with interpretation of functional connectivity from rs-fMRI. Presumably rs-fMRI samples spontaneous mental activity. It seems likely that this might include attentional-executive mechanisms but is unlikely to include 'spontaneous' reading. Thus the weak connectivity to language regions may reflect this. It might be better to use task-based connectivity and contrast the connectivity of the VWFA to each network during attention vs. reading.

Reviewer #2 (Remarks to the Author):

Over the past century and a half, human brain mapping consisted of pinning small functionally responsive areas within the brain. However, this functional mapping of the brain is mostly unimodal. Such an approach is single sighted and cannot disentangle whether a brain area responsiveness to a given function should be considered as specific to this function or an essential contributor to this function. The latter option is problematic because it challenges the fundamental reasoning of functional localisation.

Chen, Wasserman et al. challenged themselves to revisit a classically defined area– the visual word form area and demonstrate that its function goes beyond its location but rather correlates with what it is connected with.

I very much enjoyed reading the manuscript, and I found the neuroimaging methods spotless.

I have, however a few minor comments.

"Third, previous VWFA connectivity studies have restricted behavioral analyses to reading measures", not entirely true a recent study (Thiebaut de Schotten et al. Cerebral Cortex 2014) investigated the link between structural connectivity of the VWFA and strength of functional activation during spoken language.

"SMG, (-56, -43, 31)" This is a bit posterior for the SMG, very close to the angular gyrus. If this is not too much a pain, I'll be more comfortable if the authors moved their ROI a bit more anteriorly and superiorly.

"Age-adjusted standardized scores for the TORRT, PVTT, and Flanker tasks were used in brain-behaviour analyses" would it be possible to specify what kind of adjustment was employed?

"Cohen's f and Bayes Factor were calculated for the regression models to reveal the effect size and power of each model" I believe Cohen's f and Bayes Factor indicate the effect size but not the power of each model.

While the effect size is important to report, it is not necessary to report the power of the analysis as the sample is large enough to speak for itself.

The effect size of the correlations between brain and behaviour were not that strong. I was wondering whether the authors attempted to approach this brain and behaviour relationship with non-parametric correlations.

Finally, the authors should clarify whether the brain and behaviour analyses were corrected for multiple comparisons.

Reviewer #3 (Remarks to the Author):

The manuscript by Chen, Wassermann et al. investigates the nature of the visual word form area (VWFA) and the role it plays in perception/cognition. To do so, the authors take the publicly released Human Connectome Project (HCP) data, quantify structural and functional connectivity of the VWFA to other brain regions, and look for relationships between these results and behavioral measures that were also acquired on HCP participants.

As far as the results go, I do not find anything intrinsically objectionable about them; the analyses and quantifications are conventional (which is fine). However, it is not really clear what substantial scientific insights the results provide. The paper is couched and written as if there are two theories in the literature ('language' vs. 'multiplexed' models), but these theories are not mutually exclusive (the VWFA is intimately connected to both attention and language). Furthermore, I do not think that researchers in the field actually subscribe to only one of these two general theories. Thus, it appears the paper sets up somewhat of a straw-man argument. Furthermore, in terms of scientific insights, structural connectivity and resting-state functional connectivity are fine for what they are, and quantifying these measures does provide some information. However, it is not clear whether these measures, in and of themselves, provide novel insights into the nature of the VWFA. On basis of having made these quantifications in the HCP dataset, I wonder if the authors can propose a new theory (or hypothesis) about the nature of the VWFA and then execute a specific focused experiment that can shed light on some mechanism that they believe is at play.

As an example of how one might derive theoretical insight, one might ask: what is the potential mechanistic link between the diffusion quantifications made in the current manuscript and behavior? (A correlation between two measures does not reveal intervening mechanisms.) As another example, it is stated that the structural connectivity of VWFA to fronto-parietal attention-related areas is greater than the connectivity with language networks; although this is a conclusion that the analyses can establish, it is not clear if there is a meaningful conclusion to be drawn here (for example, just because one set of tracts is larger than another set of tracts, is there any conclusion to be made about the significance to cognitive functioning of those two sets of tracts?). Similar concerns affect the functional connectivity results: during rest, some brain areas appear to have activity correlations with one another, but what insight does this provide regarding the role of VWFA when subjects are reading or engaged in linguistic function?

One minor methodological comment: For defining the location of VWFA, the use of average MNI coordinates has limited accuracy. For example, the fusiform face area (FFA) typically borders the VWFA, and so the use of group atlas coordinates will likely include some non-VWFA cortex due to inter-subject variability in the location of these areas.

Comments from the Reviewer 1:

R1.1 *“Classically, the function of this vOT region was influenced by neuropsychological data (from Dejerine onwards). The Behrmann and Plaut studies are important. In addition, the largest cases series of such patients (Roberts et al., Cerebral Cortex 2012) is not considered at all. That study shows that there are co-occurring non-reading deficits across the patients including weakness with visually-complex novel and familiar stimuli including faces. Given the breadth of assessment and number of patients reported, this study and its findings need to be considered in the paper. This study sets out many reasons why reading is a highly-demanding visual decoding task (which requires rapid online visual perception, attention, eye movement programming – see Leff et al, as well as conversion to language representations)”*

We thank the reviewer for pointing out the connections of our work to the excellent neuropsychological studies by Roberts and Leff. In the revised manuscript, we have referred to these studies in the Introduction and Discussion (pp. 4, 25, 31).

R1.2 *“Vertical occipital fasciculus – it is claimed here that this fasciculus connects VWFA to IPS. Is this correct? My understanding was that this fibre bundle is much more posterior. Even in the authors’ own Figure 2, the fibres terminate posterior to the VWFA marked seed and project dorsally to the occipito-parietal junction (not to IPS).”*

We thank the reviewer for this observation and apologize for some confusion here. First, all structural connectivity analyses originated from the VWFA, and therefore, by definition, these

tracts terminate ventrally in the VWFA. While the blue tracts shown in Figure 2 appear to terminate ventrally in a locus posterior to VWFA, as noted by the reviewer, this appearance is due to the fact that the green tracts running anterior towards the temporal lobe language areas are overlaid on top of these blue tracts and obscure their actual termination point. We show the VWFA termination point for the blue tract below in Figure R1. Second, even at the extremely conservative threshold for the tracts shown in Figure 2 ($p < 10^{-15}$), the dorsal termination point for this tract is in the posterior IPS, more dorsal and anterior than indicated in previous studies^{4,5}. Using a less conservative threshold, yet more stringent than those used in previous studies ($p < 10^{-5}$), we clearly demonstrate that the dorsal termination point for this tract extends further dorsally into the IPS. Third, our ROI analysis revealed highly significant structural connectivity for this tract within both the aIPS and pIPS using *a priori* defined ROIs for these IPS target regions ($p < .001$; Figure 2a and 2e and Table S3). Fourth, since this tract originates in the VWFA and extends dorsally into the IPS, we have adopted the nomenclature (vertical occipital fasciculus, VOF) used in a previous study by Kay and Yeatman et al⁵, which identified a white matter tract originating in the VWFA and terminating in the posterior IPS in all participants. To highlight the similarity between the VOF identified in our structural connectivity analysis and the Yeatman's study⁴, we have plotted results from both of these studies in Figure R1 below. This comparison shows a striking similarity for the dorsal termination of this tract in the IPS from both our HCP dataset (left and middle) and the Yeatman (right) study. Furthermore, our results show that VWFA tracts extend more dorsally and anteriorly within the IPS than suggested by Yeatman et al. We have made every attempt to clarify these points in the revised manuscript (page 16).

Figure R1: Leftmost: The dorsal component of VWFA connectivity with IPS using deterministic method, and the blue tract (also shown in Figure 2 and S1) was considered as vOF. Middle: VWFA connectivity from the present study with a stringent threshold reported in the main text ($p < 10^{-15}$) and a less conservative threshold ($p < 10^{-5}$) showing a larger coverage of the IPS by the VWFA connections; Rightmost: vOF terminations reproduced from Yeatman et al PNAS 2014. The overlap is strongest in the posterior aspects of the IPS.

R1.3 “There is a more fundamental assumption that should be addressed. The study examines white-matter connectivity. It is likely that these structural pathways are present long before reading tuition commences and thus this neural network must reflect more basic primary visual and language functions – which reading is parasitic on. Although there is evidence that experience can modulate strength of connection and cortical volume, as far as I am aware there is no evidence that experience can generate new patterns of connectivity across the cortex.”

We thank the reviewer for these comments. It was not our intention to state that experience generates new patterns of connectivity across the cortex. We agree with the reviewer that the

structural and functional pathways between VWFA and language networks are present before reading skills are acquired during the elementary school years and are strengthened during the acquisition of reading skills. We have now clarified this in the revised manuscript (pp. 26-27).

R1.4 *“Although the HCP undertook distortion correction methods for the DWI, the rs-fMRI were collected with a single echo and various aspects of the anterior temporal lobe (ATL) are known to be missing. This will limit the range of language-semantic related regions that can be examined via the rs-fMRI data. These regions are known to be involved in reading (see Hoffman et al., PNAS 2015; Woollams et al PNAS 2017).”*

We thank the reviewer for pointing out the limitations of the fMRI protocol in the HCP dataset, which are now acknowledged in the revised Methods section (pp. 9-10). However, even with this limitation in data acquisition, we were still able to detect significant intrinsic connectivity of VWFA to multiple ATL regions (see R1.5 and Table S6).

R1.5 *“It would be useful to include anterior temporal regions in the language-related ROIs (see point above). Which is reinforced given the primary connection along the ILF.”*

As suggested by the reviewer, we have performed additional ROI analyses to examine structural and intrinsic functional connectivity between VWFA and multiple anterior temporal lobe (ATL) regions identified in the reading and language literatures. ATL ROIs included regions identified by Woollams et al⁶, Hoffman et al⁷, and Visser et al⁸. Results from both structural and intrinsic functional connectivity analyses revealed significant connectivity between VWFA and all of these ATL regions. Furthermore, brain-behavior results showed a significant correlation between the strength of VWFA-ATL intrinsic functional connectivity and picture vocabulary scores across individuals. We have included these new analyses and results, and their discussion, in the revised manuscript (pp. 18, 20, 23, 25-26, Tables S6 and S11).

R1.6 *“Tract strength was interpreted as the proportion of streamline between the VWFA and each ROI. Note that streamline numbers are influenced by many factors other than point-to-point connectivity including forking, crossing and fanning fibres. The values are also influenced by tract distance. How were these confounds dealt with?”*

We thank the reviewer for this comment. Our probabilistic tractography method is based on the constrained spherical deconvolution (CSD) deterministic tractography model⁹, which is a well-established method for accounting for crossing and fanning fibers. Moreover, we have employed established methods for correcting for tract distance¹⁰. We now specify these important methodological considerations in the revised Methods (pp. 10-11) and further acknowledge that forking fibers continue to represent a methodological challenge using current DTI techniques.

R1.7 *“A t-test against zero was used to assess the presence of a connection. I do not understand this. Surely all probably values (proportion of streamlines) will be >0 by definition. No negative values will be present. Isn't a different baseline needed? E.g., the level of connectivity between randomly-selected ROIs? There will be a minimal number of streamlines simply by chance.”*

We thank the reviewer for this thoughtful comment and have revised our analysis to address this issue. First, we would like to clarify that all group-level ROI analyses in our study were conducted on age- and gender-adjusted beta values for both the structural and the intrinsic functional and connectivity analyses. Critically, these connectivity beta values can be *either* positive or negative. However, we agree with the reviewer that a different baseline is necessary given that structural connectivity is based on the proportion of streamlines. Therefore, we have revised our analysis using thresholding methods used by Binney et al¹¹. Specifically, we first calculated the probability of all cortico-cortical connections at the vertex level for each subject, estimated the 95-percentile value of the probability of all connections for the subject, and computed the median value across subjects (0.00402). We then used this population mean as the empirical null value for our statistical analysis, and tested the hypothesis that vertex-based connectivity is significantly higher than this value. A threshold of $p < 10^{-15}$ (FDR corrected for multiple comparisons) was used for generating the group-average map in Figures 2 and R1 (pp. 10-11).

R1.8 *“There were multiple behaviour measures for language and attention. Which of these were used in the brain-behaviour regression analysis and how? Were the analyses repeated for each measure or were the measures combined in some way?”*

We thank the reviewer for this comment. In the brain-behavior regression analysis, we used a total of three behavioral measures, including two language (word reading and picture vocabulary) and one attention measure (flanker). The same regression analysis procedures were performed separately for each behavioral measure, and these behavioral measures were not combined. We have clarified these points in the revised Methods (Figure 1 figure caption, pp. 12-14).

R1.9 *“As noted above – I do not understand how “significant” connections to the ROIs is being assessed or what the “effect size” is in this context?”*

We thank the reviewer for this comment, and as we stated above in R1.7, we now test the significance of structural connections based on an empirically derived null value from all subjects. Given our large sample size, we also computed the “effect size” as the magnitude of these differences, to enable comparison of our results with previous and future research studies. We have revised our Results and Discussion (pp. 17-20, 25-27) to reflect this change.

R1.10 *“Apologies if I have missed this in the Methods section but I do not understand how the comparison of connectivity (structural or functional) between VWFA and either the language or attention networks is computed? Are the values for the different connections across people combined in some way?”*

Connectivity between VWFA and the language and attention networks was performed using a theoretically-driven, ROI-based approach. Specifically, we first computed connectivity betas for each link between VWFA and each node of the language network, using *a priori* ROIs from Vogel et al.¹² and a Neurosynth meta-analysis map for the search term “reading”. We then computed the average beta value across these links for each participant. This procedure was then performed for the links connecting VWFA to the visuospatial attention network using *a priori*

ROIs from Vogel et al.¹² and retinotopic maps of IPS. These betas, which represent the average connectivity between VWFA and the language and attention networks for each participant, were then compared using a paired-samples *t*-test. This analysis was conducted separately for intrinsic functional and structural connectivity measures. We have included a new subsection to the revised Methods entitled “*Comparison of VWFA connectivity to the language vs. attention network*” (page 12) detailing this analysis.

R1.11 “*Apologies again – but what method was use to extract the ILF, AF and vOF as shown in Figure 2?*”

We thank the reviewer for this comment. The ILF, AF and vOF were detected using constrained spherical deconvolution (CSD)-based deterministic tractography⁹. The tracts were identified based on our whole-brain structural connectivity and a literature review of known white matter fascicles innervating the ventral temporal lobe (see e.g. Wassermann et al¹³ and Yeatman et al⁴). Based on this analysis, we used regions of interest to segment the deterministic tractography results to extract tracts linking specific brain areas of the temporal (ILF), frontal (AF) and parieto-occipital (vOF) lobes. The small brain plotted in Figure 2a shows data from one representative individual. We have clarified these details in the revised manuscript (Figures 2, figure caption and Figure S1).

R1.12 “*I note that the R² for the regression models are not very high. Thus, it might be important not to overstate the brain-to-behaviour links. The connectivity-behavioural correlations show some limited relationship but the vast majority of the behavioural variance is left unexplained.*”

We agree with the reviewers that the R^2 values of the regression models are modest, and we have made every effort not to overstate the brain-behavior links while highlighting these relationships. In the revised manuscript we acknowledge that further data-driven (as opposed to hypothesis-driven) research is needed to identify sources of additional behavioral variance (page 23).

R1.13 “*It is probably also important to be careful with interpretation of functional connectivity from rs-fMRI. Presumably rs-fMRI samples spontaneous mental activity. It seems likely that this might include attentional-executive mechanisms but is unlike to include ‘spontaneous’ reading. Thus the weak connectivity to language regions may reflect this. It might be better to use task-based connectivity and contrast the connectivity of the VWFA to each network during attention vs. reading.*”

We thank the reviewer for this comment regarding interpretability of functional connectivity based on rs-fMRI. Accordingly, we have added new analyses to the revised manuscript to examine task-related functional engagement and connectivity during reading and attention tasks in VWFA circuits. Importantly, task-based results show strong convergence with the previously reported rs-fMRI results: our new results reveal task-based functional connectivity between VWFA and nodes of the language and attention networks during word reading, and task-based functional connectivity between VWFA and nodes of the attention network during visuospatial attention. Moreover, these intrinsic and task-based functional results show a strong convergence with structural connectivity results, which are not influenced by attentional-executive processes

that may be engaged in the scanner. Our results demonstrate that task-based functional connectivity shows strong convergence with findings from structural and intrinsic functional connectivity, and highlight a tight link between intrinsic VWFA circuits and cortical function^{14,15}.

Comments from Reviewer 2:

R2.1 “ ‘Previous VWFA connectivity studies have restricted behavioral analyses to reading measures’, *not entirely true a recent study (Thiebaut de Schotten et al. Cerebral Cortex 2014) investigated the link between structural connectivity of the VWFA and strength of functional activation during spoken language.*”

We thank the reviewer for pointing us to this important work and we have added this reference to our revised Introduction (*page 5*).

R2.2 “*SMG, (-56, -43, 31)...is a bit posterior for the supramarginal gyrus (SMG), very close to the angular gyrus. If this is not too much a pain, I'll be more comfortable if the authors moved their ROI a bit more anteriorly and superiorly.*”

We thank the reviewer for the suggestion to include a more anterior and superior ROI for the supramarginal gyrus. Using the Neurosynth activation map for the key word term “reading” confined with the Harvard-Oxford atlas, we chose a more anterior peak for the SMG (-44, -38, 40) to conduct the same analysis and we found very similar results, and we have included these results in the main text (*pp. 9, 17*) and Table S7.

R2.3 “ ‘Age-adjusted standardized scores for the TORRT, PVT, and Flanker tasks were used in brain- behaviour analyses’ *would it be possible to specify what kind of adjustment was employed?*”

We thank the reviewer for raising this this important question. All behavioral measures included in the HCP dataset, and used in our study, were from NIH Toolbox. Adjustments for age were computed based on a national norm (http://www.healthmeasures.net/images/nihtoolbox/Training-Admin-Scoring_Manuals/NIH_Toolbox_Scoring_and_Interpretation_Manual_9-27-12.pdf). To clarify these points, we have included the following in the revised Methods (*page 13*): “For adults, age bands are used (18-29, 30-39, 40-49, 50-59, 60-69, 70-79, 80-85), following generally accepted practices in norm-referenced test development. A score of 100 indicates performance that was at the national average for the test-taking participant’s age. A score of 115 or 85, for example, would indicate that the participant’s performance is 1 SD above or below the national average, respectively, when compared with like-aged participants. Higher scores indicate better performance.” Based on this description, the age-adjusted scores were normalized according to the mean and std. deviation of each age group.

R2.4 “ ‘Cohen’s *f* and Bayes Factor were calculated for the regression models to reveal the effect size and power of each model’ *I believe Cohen’s f and Bayes Factor indicate the effect size but not the power of each model. While the effect size is important to report, it is not necessary to report the power of the analysis as the sample is large enough to speak for itself.*”

We thank the reviewer for this comment. We would like to provide the Bayes Factor measures, which capture the strength of evidence, to complement conventional significance testing. We agree with the reviewer that the large sample size speaks to the power of the analysis and have removed it from the revised manuscript.

R2.5 “*The effect size of the correlations between brain and behaviour were not that strong. I was wondering whether the authors attempted to approach this brain and behaviour relationship with non-parametric correlations.*”

We thank the reviewer for suggesting a non-parametric approach for examining brain-behavior relationships. Accordingly, in addition to the parametric linear regression analysis (Figure 4), we have added a general additive model (GAM) analysis, which incorporates cubic regression spline¹⁶ to examine brain-behavior relationships. We also compared the model fit of the parametric linear model with the GAM on the same measures and used Spearman correlations to examine the relationship between the predicted values and the actual values on each behavioral measure. Overall, results from non-parametric analysis revealed similar results, and provided only a subtle advantage relative to parametric results, with the proportion of variance explained increasing by 1.7%~6.8%. We have summarized these results in the revised manuscript (page 22 and Table S14 and S15).

R2.6 “*Finally, the authors should clarify whether the brain and behaviour analyses were corrected for multiple comparisons.*”

We thank the reviewer’s comments regarding corrections for multiple comparisons in the brain-behavior analysis. In one component of the brain-behavior analyses, we examine *a priori* hypotheses regarding VWFA connectivity. Specifically, we examine whether (a) connectivity between VWFA and nodes of the language network predict the two language measures (word reading and picture vocabulary), and (b) connectivity between VWFA and nodes of the visuospatial attention network predict the attention measure (flanker). Given our *a priori* hypotheses regarding these aspects of VWFA connectivity, these analyses were performed with FDR correction and results were significant at the $p < .05$ level. A second component of the brain-behavior analyses examined the specificity of these initial finding. Specifically, these latter analyses examined whether: (a) connectivity between VWFA and language nodes predicted flanker scores, and (b) connectivity between VWFA and attention nodes predicted language scores. We did not require corrections for multiple comparisons for this second aspect of the brain-behavior analysis to enable a more stringent test of specificity for the initial set of results. We have clarified these important points in the revised manuscript (Methods, page 15), and have provided both the corrected and uncorrected p-values for all analyses in the revised Results (page 23) and in Figure 4.

Comments from Reviewer 3:

R3.1 “*As far as the results go, I do not find anything intrinsically objectionable about them; the*

analyses and quantifications are conventional (which is fine). However, it is not really clear what substantial scientific insights the results provide. The paper is couched and written as if there are two theories in the literature ('language' vs. 'multiplexed' models), but these theories are not mutually exclusive (the VWFA is intimately connected to both attention and language). Furthermore, I do not think that researchers in the field actually subscribe to only one of these two general theories. Thus, it appears the paper sets up somewhat of a straw-man argument."

We thank the reviewer for their critical insight regarding theories of VWFA function. While we share many of the reviewer's perspectives, a close reading of the literature reveals a strong language focus for the VWFA in many works, with limited acknowledgement of a role for VWFA in attention and its related circuitry. For example, in a paragraph highlighting the importance of examining VWFA connectivity to language regions, a recent high-impact publication¹⁷ examining VWFA connectivity states:

"Even more troubling, other studies have reported preferential RSFC between the VWFA and brain regions associated with visual attention, rather than language (Vogel et al., 2012a; Zhou et al., 2015). These latter findings are more in keeping with claims that the VWFA is nothing more than a general visual processor for discriminating high-spatial- frequency stimuli of any kind (Price and Devlin, 2003, 2004; Vogel et al., 2012b)."

Moreover, a recent review on VWFA function¹⁸ does not mention the word "attention" throughout the review, focuses primarily on regional activation in VWFA, and does not discuss a role for parietal visuo-spatial and attentional brain systems, or their interconnected circuitry. Our goal in framing the present study was to acknowledge the strong emphasis of the VWFA's role in language function in some of these previous works, and to go beyond local activation profiles of VWFA to examine intrinsic, structural, and task-based connectivity patterns of VWFA and their relation to reading and attentional abilities. We believe that such a multimodal approach has been missing in the literature and provides the most convincing proof yet that the VWFA is intimately connected to both attention and language networks, and, furthermore, that VWFA connectivity patterns are behaviorally dissociable.

R3.2 *"Furthermore, in terms of scientific insights, structural connectivity and resting-state functional connectivity are fine for what they are and quantifying these measures does provide some information. However, it is not clear whether these measures, in and of themselves, provide novel insights into the nature of the VWFA. On basis of having made these quantifications in the HCP dataset, I wonder if the authors can propose a new theory (or hypothesis) about the nature of the VWFA and **then execute a specific focused experiment** that can shed light on some mechanism that they believe is at play."*

We thank the reviewer for this comment and agree that structural and intrinsic connectivity measures are somewhat limited in their ability to provide insights into functional brain networks in isolation. While initiating a new fMRI task experiments to test novel hypotheses of VWFA function was beyond the scope of the current work, in the revised manuscript we followed the suggestions from the Editor and Reviewer 1 to extend our results to task fMRI data. Specifically,

we now report analysis of task-based functional connectivity during reading and attention tasks using the same seeds and regions of interest (ROI) described in the initial manuscript submission. Consistent with our findings from structural and intrinsic connectivity in the initial submission, our new task-based connectivity results strongly support the multiplex model of VWFA function^{1,2,5,12}.

R3.3 *“As an example of how one might derive theoretical insight, one might ask: what is the potential mechanistic link between the diffusion quantifications made in the current manuscript and behavior? (A correlation between two measures does not reveal intervening mechanisms.) As another example, it is stated that the structural connectivity of VWFA to fronto-parietal attention-related areas is greater than the connectivity with language networks; although this is a conclusion that the analyses can establish, it is not clear if there is a meaningful conclusion to be drawn here (for example, just because one set of tracts is larger than another set of tracts, is there any conclusion to be made about the significance to cognitive functioning of those two sets of tracts?). Similar concerns affect the functional connectivity results: during rest, some brain areas appear to have activity correlations with one another, but what insight does this provide regarding the role of VWFA when subjects are reading or engaged in linguistic function?”*

We thank the reviewer for these thoughtful comments. While we are hesitant to make a strong claim regarding the finding showing greater VWFA connectivity to the frontoparietal network compared to the language nodes, we have stated that the VWFA shows strong structural and functional connectivity with both the frontoparietal attention and temporal lobe language-related areas. Furthermore, VWFA connectivity patterns to these networks are behaviorally dissociable. The theoretical insight of these findings is that the VWFA is well placed to integrate signals from the attention and language systems.

R3.4 *“Similar concerns affect the functional connectivity results: during rest, some brain areas appear to have activity correlations with one another, but what insight does this provide regarding the role of VWFA when subjects are reading or engaged in linguistic function?”*

We agree with the reviewer that functional task data is required to further clarify the link between VWFA connectivity and reading and attention function. We have now added new data from two fMRI tasks. In response to this suggestion, we analyzed two OpenNeuro task fMRI datasets, including word reading/rhyming judgment (<https://openneuro.org/datasets/ds000003/versions/00001>) and visuospatial attention/flanker tasks (<https://openneuro.org/datasets/ds000102/versions/00001>). We used the generalized psychophysiological interaction (gPPI) model³ to examine task-based functional connectivity and employed the same seeds and regions of interest (ROI) described in the structural and intrinsic functional connectivity analyses from the initial manuscript submission. Results from task-based functional connectivity analyses reveal that: (1) the VWFA was strongly activated during both word reading and flanker attention tasks; (2) VWFA task-based functional connectivity increased to: (a) language network nodes, including IFG, STS, as well as attention network nodes, including FEF and IPS, during the reading task and (b) visuospatial attention network nodes during the visuospatial attention tasks. ***Importantly, task-based functional connectivity results show strong convergence with our structural and intrinsic functional connectivity results.***

R3.5 *“For defining the location of VWFA, the use of average MNI coordinates has limited accuracy. For example, the fusiform face area (FFA) typically borders the VWFA, and so the use of group atlas coordinates will likely include some non-VWFA cortex due to inter-subject variability in the location of these areas.”*

We agree with this reviewer comment and believe that the use of group atlas coordinates should make it more difficult to find consistent patterns of connectivity with distal brain regions relative to participant-specific coordinates, such as those used by Stevens et al.¹⁷. Despite this limitation, our analyses identified a significant relationship between reading abilities and the strength of functional connectivity between VWFA and nodes of the language network. The replicability of results across methodologies and ROIs from previous studies strongly suggest the importance of these links for language and attention. Nevertheless, we acknowledge that using individualized ROIs using localizer tasks might increase effect sizes in brain-behavioral correlations. Unfortunately, the HCP dataset did not include localizer fMRI tasks.

Reference

1. Behrmann, M. & Plaut, D. C. Distributed circuits, not circumscribed centers, mediate visual recognition. *Trends Cogn. Sci.* **17**, 210–219 (2013).
2. Vogel, A. C., Petersen, S. E. & Schlaggar, B. L. The VWFA: it's not just for words anymore. *Front. Hum. Neurosci.* **8**, 88 (2014).
3. McLaren, D. G., Ries, M. L., Xu, G. & Johnson, S. C. A generalized form of context-dependent psychophysiological interactions (gPPI): A comparison to standard approaches. *Neuroimage* (2012). doi:10.1016/j.neuroimage.2012.03.068
4. Yeatman, J. D. *et al.* The vertical occipital fasciculus: A century of controversy resolved by in vivo measurements. *Proc. Natl. Acad. Sci.* 201418503 (2014). doi:10.1073/pnas.1418503111
5. Kay, K. N. & Yeatman, J. D. Bottom-up and top-down computations in word- and face-selective cortex. *Elife* **6**, 1–29 (2017).
6. Woollams, A. M., Madrid, G. & Lambon Ralph, M. A. Using neurostimulation to understand the impact of pre-morbid individual differences on post-lesion outcomes. *Proc. Natl. Acad. Sci. U. S. A.* **114**, 12279–12284 (2017).
7. Hoffman, P., Binney, R. J. & Lambon Ralph, M. A. Differing contributions of inferior prefrontal and anterior temporal cortex to concrete and abstract conceptual knowledge. *Cortex* **63**, 250–266 (2015).
8. Visser, M. & Lambon Ralph, M. A. Differential contributions of bilateral ventral anterior temporal lobe and left anterior superior temporal gyrus to semantic processes. *J. Cogn. Neurosci.* **23**, 3121–3131 (2011).
9. Jeurissen, B., Leemans, A., Jones, D. K., Tournier, J. D. & Sijbers, J. Probabilistic fiber tracking using the residual bootstrap with constrained spherical deconvolution. *Hum. Brain Mapp.* **32**, 461–479 (2011).
10. Anwander, A., Tittgemeyer, M., Von Cramon, D. Y., Friederici, A. D. & Knösche, T. R. Connectivity-based parcellation of Broca's area. *Cereb. Cortex* (2007). doi:10.1093/cercor/bhk034
11. Binney, R. J., Embleton, K. V., Jefferies, E., Parker, G. J. M. & Lambon Ralph, M. A. The ventral and inferolateral aspects of the anterior temporal lobe are crucial in semantic memory: evidence from a novel direct comparison of distortion-corrected fMRI, rTMS, and semantic dementia. *Cereb. Cortex* **20**, 2728–38 (2010).
12. Vogel, A. C., Miezin, F. M., Petersen, S. E. & Schlaggar, B. L. The putative visual word form area is functionally connected to the dorsal attention network. *Cereb. Cortex* **22**, 537–549 (2012).
13. Wassermann, D. *et al.* The white matter query language: a novel approach for describing human white matter anatomy. *Brain Struct. Funct.* 1–17 (2016).
14. Cole, M. W., Ito, T., Bassett, D. S. & Schultz, D. H. Activity flow over resting-state networks shapes cognitive task activations. *Nat. Neurosci.* (2016). doi:10.1038/nn.4406
15. Cole, M. W. *et al.* Multi-task connectivity reveals flexible hubs for adaptive task control. *Nat. Neurosci.* (2013). doi:10.1038/nn.3470
16. Chiang, A. Y. Generalized Additive Models: An Introduction With R. *Technometrics* (2007). doi:10.1198/tech.2007.s505
17. Stevens, W. D., Kravitz, D. J., Peng, C. S., Henry Tessler, M. & Martin, A. Privileged Functional Connectivity Between the Visual Word Form Area and the Language System.

- J. Neurosci.* **37**, 0138–17 (2017).
18. Hannagan, T., Amedi, A., Cohen, L., Dehaene-Lambertz, G. & Dehaene, S. Origins of the specialization for letters and numbers in ventral occipitotemporal cortex. *Trends Cogn. Sci.* **19**, 374–382 (2015).

REVIEWERS' COMMENTS:

Reviewer #1 (Remarks to the Author):

The authors have completed a substantial and considered set of thorough revisions. I believe they have addressed all of the comments from all three reviews, including myself. Whilst it would be good to undertake directed, new fMRI explorations of reading and attention in the same participants to test the hypotheses of this study in more detail, the use of open access data are a good first step.

Reviewer #2 (Remarks to the Author):

I am delighted by the revision of the manuscript, and the authors did an excellent job. However, it appears to me that the vertical occipital fasciculus (VOF) described by the authors is not occipital but rather temporo-parietal. Further, the projections do not correspond to the cortical projections reported by Oishi et al. (PNAS 2018 - figure 3) I wonder whether this track is the posterior segment of the arcuate fasciculus (Catani et al. 2005) instead of the VOF.

I am conscious that the anatomy of VOF has been confusing for a long time, but the team of Hiromasa Takemura has recently clarified the anatomy of the VOF, which as the name indicate should be an occipital tract. Anatomy matters to me a lot, and I was wondering whether the authors would have the kindness to amend their manuscript with the correct tract name for clarity and replicability of their findings.

The rest of the manuscript is outstanding.

Reviewer #3 (Remarks to the Author):

I think the addition of the analyses of the OpenNeuro datasets provides value to the paper. And it is certainly re-assuring to find high activity and correlated activity in VWFA and language regions during a reading task, and, conversely, high activity and correlated activity in VWFA and attention-related regions during a visuospatial task.

However, I feel that the treatment of these task-based data is still conceptually superficial and does not provide substantial new insight into how the VWFA engages in its language and attention-related capacities. Thus, I feel that the fundamental concern I raised in my initial review remains - that the results of the paper do not provide substantial novel theoretical insight into the nature of the VWFA.

Response to reviewers

Reviewer 2

R.2.1: *“it appears to me that the vertical occipital fasciculus (VOF) described by the authors is not occipital but rather temporo-parietal. Further, the projections do not correspond to the cortical projections reported by Oishi et al. (PNAS 2018 - figure 3) I wonder whether this track is the posterior segment of the arcuate fasciculus (Catani et al. 2005) instead of the VOF. I am conscious that the anatomy of VOF has been confusing for a long time, but the team of Hiromasa Takemura has recently clarified the anatomy of the VOF, which as the name indicate should be an occipital tract. Anatomy matters to me a lot, and I was wondering whether the authors would have the kindness to amend their manuscript with the correct tract name for clarity and replicability of their findings.”*

We thank the reviewer for this comment, and we agree that the structural neuroanatomy literature has produced some uncertainty regarding the nomenclature of this white matter tract. To address this issue, we have highlighted the uncertainty of the nomenclature in the revised Results and relabeled the tract as vOF/TP-SPL throughout the manuscript.